# Quantifying the impact of COVID-19 on immigration in receiving high-income countries

Miguel González-Leonardo[1]*, Michaela Potančoková[1], Dilek Yildiz[1,2], Francisco Rowe[3]

**1** International Institute for Applied Systems Analysis, Wittgenstein Centre for Demography and Global Human Capital (IIASA, VID/OEAW, University of Vienna), Laxenburg, Austria, **2** Vienna Institute of Demography/Austrian Academy of Sciences, Wittgenstein Centre for Demography and Global Human Capital (IIASA, VID/OEAW, University of Vienna), Vienna, Austria, **3** Geographic Data Science Lab, Department of Geography and Planning, University of Liverpool, Liverpool, United Kingdom

* gonzalezm@iiasa.ac.at

**Data Availability Statement:** All relevant data are available at Github: https://github.com/MiguelGonzalezLeonardo/The-impact-of-COVID-19-on-immigration.

## Abstract

Previous studies have examined the impact of COVID-19 on mortality and fertility. However, little is known about the effect of the pandemic on constraining international migration. We use Eurostat and national statistics data on immigration and ARIMA time-series models to quantify the impact of COVID-19 on immigration flows in 15 high-income countries by forecasting their counterfactual levels in 2020, assuming no pandemic, and comparing these estimates with observed immigration counts. We then explore potential driving forces, such as stringency measures and increases in unemployment moderating the extent of immigration change. Our results show that immigration declined in all countries, except in Finland. Yet, significant cross-national variations exist. Australia (60%), Spain (45%) and Sweden (36%) display the largest declines, while immigration decreased by between 15% and 30% in seven countries, and by less than 15% in four nations where results were not statistically significant. International travel restrictions, mobility restrictions and stay-at-home requirements exhibit a relatively strong relationship with declines in immigration, although countries with similar levels of stringency witnessed varying levels of immigration decline. Work and school closings and unemployment show no relationship with changes in immigration.

## Introduction

The COVID-19 pandemic has had major impacts on global demographic trends. Research has measured reductions in life expectancy [1–3] and fertility resulting from the pandemic [4, 5], emphasizing that significant cross-national differences exist. However, less is known about the impact of the pandemic on international migration. Despite some evidence pointing to a reduction in this component [6, 7], the extent of this decline and variations across countries are yet to be established. Lack of timely data has prevented us from quantifying these trends, as information on international migration flows during 2020 has recently become available only for some high-income countries.

**Funding:** The authors gratefully acknowledge funding from IIASA and the National Member Organizations that support the institute. This analysis was supported by European Union's Horizon 2020 research and innovation programme projects QUANTMIG (870299) & FUME (870649).

**Competing interests:** The authors have declared that no competing interests exist.

International migration is the primary component of population change in aging societies. Net migration gains have been identified as a key demographic process to prevent or at least mitigate depopulation and labor force deficits [8, 9]. Inflows of international migration increase the number of young adults and elevate fertility [10, 11]. They also bring labor force and skills where they are needed [12] and support the welfare state and intergenerational transfers by sustaining suitable labor dependency ratios [8]. Thus, monitoring movements between countries and understanding changes in the global network of international migration is essential to ensure appropriate policies in countries where natural change (difference between births and deaths) cannot sustain population growth and where labor force deficits exist.

Human mobility and the spread of infectious diseases are closely related [13]. Several studies have demonstrated that human mobility patterns have influenced the ways in which COVID-19 has spread both across and within countries [14–16]. National governments rapidly implemented mobility restrictions and border closures, among other stringency measures, to contain the spread of COVID-19 at the onset of the pandemic. Consequently, one of the major early impacts of the pandemic was the increase in involuntary immobility [17]. Both levels and patterns of international migration are likely to have been significantly affected: individuals with plans to migrate were not able leave their countries, others found themselves trapped in the migration journey or struggled to return to their home countries [6, 18].

Among the major receiving countries, immigration declines may have been more pronounced where there have been greater international travel restrictions (e.g., Australia and Canada) and other stringency measures, such as strict lockdowns and mobility restrictions (e.g., Italy). Additionally, it is expected that inflows to destinations traditionally receiving sizeable volumes of immigrants from large distances and overseas (e.g., Australia and Spain) have been more impacted, as travel restrictions mostly affected air travel [18]. In sum, the closure of business and economic downturn caused by the pandemic decreased labor demand and increased unemployment in some countries [19], constraining the potential need for international workers. Thus, country-specific variations in immigration levels are expected. To date, empirical studies estimating the impact of COVID-19 on international migration are surprisingly scarce. The extent to how much immigration declined during 2020 has been documented only in Spain, estimating a reduction of almost 40% [20]. Therefore, cross-national variations on immigration due to COVID-19 are yet to be established.

We quantify cross-national impacts of COVID-19 on immigration flows in 12 European countries (Germany, Austria, France, Ireland, the Netherlands, Switzerland, Denmark, Sweden, Norway, Finland, Italy and Spain) and three non-European countries (the United States, Canada and Australia). We estimate the counterfactual level of immigration during 2020 in the absence of the pandemic, using Eurostat and national statistics data of immigration flows and Auto Regressive Integrated Moving Average (ARIMA) time-series forecasting models, and compare this level to observed counts in the same year. We also seek to identify the association between stringency measures and unemployment with immigration declines. We aim to address the following research questions: 1) To what extent did immigration decline across countries? 2) How does the extent of declines relate to stringency measures and changes in unemployment?

## Materials and methods

We used a two-stage methodology. First, we collected official statistics on immigration flows and used ARIMA models to quantify immigration declines across counties during 2020. To this end, we forecasted the counterfactual level of immigration in 2020 in the absence of the pandemic and compared this level to observed counts in the same year. Second, we used data

from Government Response Tracker [21], Eurostat and the World Bank to explore potential associations between the severity of various stringency measures and rising unemployment levels with immigration declines across countries. We tested associations using a heatmap.

## Stage 1. ARIMA models to estimate immigration declines

We used annual official statistics on immigration flows from Eurostat online database (MIGR_IMM8) between 2012 and 2020 for the European countries, from the offices of national statistics for Australia (Austrian Bureau of Statistics) and Canada (Statistics Canada), and from the Census Bureau for the United States. We restricted our analysis to immigration because of high levels of underreporting in emigration [22]. Immigrants are defined as individuals who lived over 1 year in the destination country, except for the United States, where immigration corresponds to the Census Bureau Vintage 2020 foreign-born immigration estimates.

To quantify declines in immigration, we employed country-specific ARIMA models to forecast the expected immigration counts in 2020 if the pandemic had not occurred. We then compared the forecasted immigration count to the observed immigration count in 2020. Observed counts excluded from estimated 95% confidence intervals (CIs) for predicted counts are considered statistically significant. We used 2012–2019 data to produce country-specific forecasts of immigration count for 2020. An ARIMA model comprises three components: an autoregressive (AR) process, a moving average (MA) and an integrated (I) element. Intuitively, these components capture the long-term, stochastic and short-term trends of a time series, respectively. Formally, the AR and MA components control for temporal autocorrelation in a time series resulting from two mechanisms. The first assumes a variable ($Y$) at time $t$ $(Y_t)$ which is explained by its past value(s) (i.e., $y_{t-1}, y_{t-2}, \cdots, y_{t-p}$). The second assumes $Y_t$ is a function of current and past moving averages of error terms (e.g., $u_{t-1} + u_{t-2} + \cdots + u_{t-q}$); that is, current deviations from the mean depends on previous deviations. A general ARMA ($p$, $q$) model takes the form of:

$$Y_t = \gamma + \alpha_1 Y_{t-1} + \cdots + \alpha_p Y_{t-p} - \theta_1 u_{t-1} - \cdots - \theta_q u_{t-q} + u_t \qquad (1)$$

The subscript $p$ and $q$ denote the order of the autoregressive and moving average terms, respectively. Fitting a time series in a model containing AR and MA parameters (or an ARMA model) requires the data to be weakly stationary. Weakly stationary is characterized by: (1) constant mean and variance of $Y_t$ over time; and (2) the covariance of $Y_t$ to be time-invariant, i.e., to only depend on the lag between the current and past value and not the actual time at which the covariance is computed [23]. However, weak stationarity in time series is rare. They often must be integrated (I); that is, time series must be differentiated to be stationarity so its statistical properties, such as mean, variance and autocorrelation, are constant over time. Mathematically, Eq (1) can be modified to represent a general ARIMA ($p$, $d$, $q$) model:

$$y_t = \theta + \varphi_1 y_{t-1} + \cdots + \varphi_p y_{t-p} - \beta_1 u_{t-1} - \cdots - \beta_q u_{t-q} + u_t \qquad (2)$$

where: $\boldsymbol{y_t} = Y_t - Y_{t-1}$ for a first order differencing model, and d denotes the degree of first differencing.

We fitted country-specific ARIMA models based on a combination of model selecting tools which allows us to identify the model that best fits each trend. We identified the best fitting ARIMA model for each country using unit root tests to assess for stationarity and the Akaike Information Criterion to determine the appropriate order of autoregressive, moving average and differencing terms. Models were estimated using maximum likelihood. Through our evaluation, we determined the three best fitting model specifications as shown in Table 1. Finally,

**Table 1. ARIMA model specification for each country.**

| Order of autoregressive (p), moving average (d) & differencing terms (q) | Model specification | Countries |
|---|---|---|
| p = 0; d = 0; q = 0 | White Noise model | Austria, Denmark, Germany, Finland, Italy, Sweden, Canada, the United States |
| p = 0; d = 1; q = 0 | Random Walk with a drift | France, Ireland, the Netherlands, Norway, Switzerland, Australia |
| p = 0; d = 2; q = 0 | Random Walk with a drift | Spain |

to check the robustness of our modeling strategy, we performed a sensitivity analysis by forecasting 2019 and compared it with observed values for the same year (see Sensitivity analysis in the Supporting Information (S1 Appendix)).

## Stage 2. Stringency measures and unemployment

We explored the potential relationship of immigration reductions with COVID-19 stringency levels and unemployment changes during 2020 using a heatmap. Data on stringency measures were obtained from the Oxford COVID-19 Government Response Tracker. We calculated changes in unemployment between 2019 and 2020 using data from Eurostat for European countries and from the World Bank for non-European countries.

We selected various stringency variables to analyze the association between declines in immigration and the level and type of stringency measures, including a stringency index, travel restrictions, mobility restrictions, stay-at-home requirements, work closing and school closing. The stringency index is a composite indicator that summaries the joining effect of nine individual stringency measures: school closing, workplace closing, cancelling public events, restrictions on gathering, closing of public transport, stay-at-home requirements, restrictions on internal travel, mobility restrictions and public information campaigns. The original values of this variable vary from 0 (no restrictions) to 100 (the strictest levels of restrictions).

All the individual measures of stringency we used were originally ordinal categorical variables. Travel restrictions has five categories: 0 (no restrictions), 1 (screening arrivals), 2 (quarantine arrivals from some or all regions), 3 (ban of arrivals from some regions) and 4 (full border closure). Mobility restrictions present the following values: 0 (no restrictions), 1 (recommendation of not to travel) and 2 (prohibiting internal movements). Stay-at-home can be 0 (no restrictions), 1 (recommendation of not leaving home), 2 (require not leaving house with exceptions) and 3 (total confinement with minimal exceptions). Work closing has four categories: 0 (no restrictions), 1 (workplaces can open under sanitation and social distancing requirements), 2 (closing or work from home for some sectors) and 3 (work from home and closure of non-essential activities). School closing also has three levels: 0 (no restrictions), 1 (hybrid in-person/online learning models), 2 (classes being open only for some groups), 3 (all levels of education are closed).

Stringency measures are provided as daily time series. To obtain a comparable summary indicator to our estimate of immigration decline, we calculated the annual mean for each stringency measure in 2020 based on the original data. Then we scaled the resulting means from 0 to 1, with 1 indicating the highest level in our sample. Scores for other countries are, thus, relative to the maximum record. We converted the data to this scale because a homogeneous scale for all variables was needed to produce a heatmap. See equivalences with the original values in Table 2.

**Table 2. Equivalences between Oxford stringency measures and scale 0 to 1.** Countries are ordered according to the relative change in immigration represented in brackets.

| Country | Stringency index | | Travel restrictions | | Mobility restrictions | | Stay at home | | Work School | | School closing | | Increase of unemployment | |
|---|---|---|---|---|---|---|---|---|---|---|---|---|---|---|
| | *Oxford* | *0 to 1* | *Oxford* | *0 to 1* | *Oxford* | *0 to 1* | *Oxford* | *0 to 1* | *Oxford* | *to 1* | *Oxford* | *0 to 1* | *Oxford* | *0 to 1* |
| **Australia (-59.9%)** | 55.9 | 0.9 | 3.5 | 1.0 | 1.6 | 1.0 | 1.2 | 0.7 | 1.6 | 0.8 | 1.6 | 0.6 | 25.0 | 0.2 |
| **Spain (-45.4%)** | 56.3 | 0.9 | 2.7 | 0.8 | 1.3 | 0.8 | 1.2 | 0.7 | 1.7 | 0.8 | 1.7 | 0.7 | 10.1 | 0.1 |
| **Sweden (-36.3%)** | 49.0 | 0.8 | 2.5 | 0.7 | 0.4 | 0.2 | 0.8 | 0.5 | 0.9 | 0.5 | 1.3 | 0.5 | 26.2 | 0.2 |
| **USA (-27.2%)** | 56.2 | 0.9 | 2.7 | 0.8 | 1.6 | 1.0 | 1.3 | 0.8 | 1.9 | 0.9 | 2.5 | 1.0 | 120.2 | 1.0 |
| **France (-26.5%)** | 54.3 | 0.8 | 2.5 | 0.7 | 1.1 | 0.7 | 0.8 | 0.5 | 1.7 | 0.8 | 1.6 | 0.6 | -6.2 | 0.0 |
| **Norway (-25.5%)** | 41.7 | 0.6 | 2.3 | 0.6 | 0.8 | 0.5 | 0.2 | 0.1 | 1.2 | 0.6 | 1.1 | 0.4 | 21.2 | 0.2 |
| **Germany (-21.9%)** | 51.8 | 0.8 | 2.6 | 0.7 | 1.1 | 0.7 | 0.6 | 0.3 | 1.6 | 0.8 | 1.6 | 0.6 | 24.1 | 0.2 |
| **Italy (-21.6%)** | 64.7 | 1.0 | 2.8 | 0.8 | 1.6 | 1.0 | 1.7 | 1.0 | 2.0 | 1.0 | 2.2 | 0.9 | -5.1 | 0.0 |
| **Canada (-20.2%)** | 55.8 | 0.9 | 3.3 | 0.9 | 1.6 | 1.0 | 1.2 | 0.7 | 2.0 | 1.0 | 2.2 | 0.9 | 66.7 | 0.6 |
| **Netherlands (-15.5%)** | 49.2 | 0.8 | 2.4 | 0.7 | 0.5 | 0.3 | 0.9 | 0.5 | 1.7 | 0.9 | 1.3 | 0.5 | 7.7 | 0.1 |
| **Denmark (-13.7%)** | 45.6 | 0.7 | 2.7 | 0.8 | 0.2 | 0.1 | 0.8 | 0.4 | 1.4 | 0.7 | 1.3 | 0.5 | 10.6 | 0.1 |
| **Ireland (-13.3%)** | 56.0 | 0.9 | 1.5 | 0.4 | 1.1 | 0.7 | 1.0 | 0.6 | 1.8 | 0.9 | 1.8 | 0.7 | 19.6 | 0.2 |
| **Austria (-11.1%)** | 47.1 | 0.7 | 2.4 | 0.7 | 0.6 | 0.4 | 0.7 | 0.4 | 1.5 | 0.8 | 1.2 | 0.5 | 25.5 | 0.2 |
| **Switzerland (-4.4%)** | 42.3 | 0.7 | 2.4 | 0.7 | 0.2 | 0.1 | 0.5 | 0.3 | 1.6 | 0.8 | 0.8 | 0.3 | 11.6 | 0.1 |
| **Finland (3.6%)** | 38.9 | 0.6 | 2.9 | 0.8 | 0.3 | 0.2 | 0.3 | 0.2 | 1.1 | 0.6 | 1.0 | 0.4 | 16.4 | 0.1 |

# Results

## Different immigration declines across countries

Fig 1 shows observed immigration flows between 2012 and 2020 and forecasted values for 2020 assuming continuation of observed historical trends if COVID-19 had not occurred. Fig 2 reports the percentage difference between observed and forecasted immigration counts in 2020. We consider changes in immigration as statistically significant when the observed immigration counts are outside the CI of their respective forecast. Overall, the results reveal lower than expected levels of immigration in 2020 for 14 of the 15 countries in our sample. Yet, pronounced variations exist across countries.

Australia stands out with the largest drop in immigration. The observed number of immigrants (243 thousand) was 59.9% lower than expected (607 thousand). Spain and Sweden recorded drops of 45.4% and 36.4%, declining the number of immigrants from 857 to 468 thousand and from 129 to 82 thousand, respectively. Reductions from 16% to 27% are estimated in the United States (27.2%), France (26.5%), Norway (25.5%), Germany (21.9%), Italy (21.6%), Canada (20.2%) and the Netherlands (15.5%), although results are not statistically significant in Germany due to high levels of uncertainty in the forecast. Non-statistically significant declines between 4% and 15% in Denmark, Ireland, Austria and Switzerland are more aligned with recent historical trends. In these countries, the observed drops may reflect the effect of COVID-19 on immigration, but disentangling these impacts is challenging due to uncertainty levels forecasting immigration (see Sensitivity analysis in the SI). Surprisingly, Finland recorded a slightly higher than expected immigration flow, albeit statistically insignificant.

## Association with stringency measures and unemployment change

Next, we examine the association between reductions in immigration with stringency measures and unemployment changes. We selected five indicators capturing specific restrictions including travel, mobility, stay-at-home, work and school closures, and a stringency

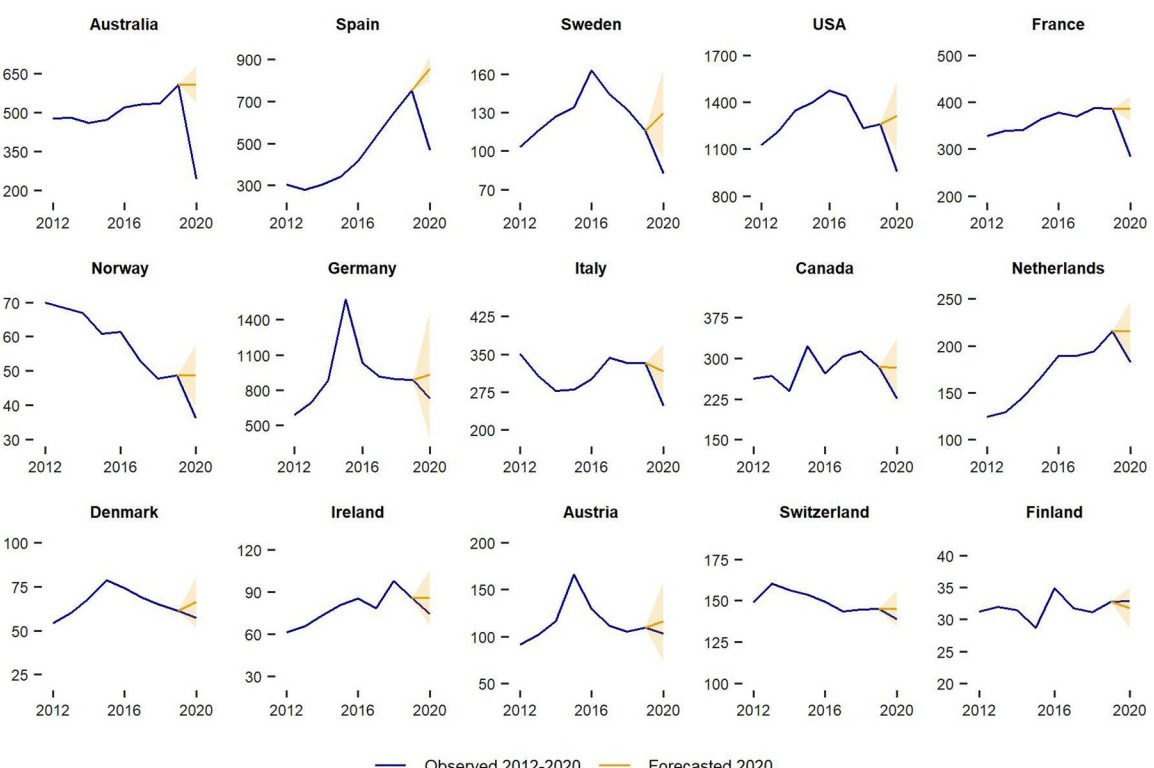

**Fig 1. Immigration (thousands): Observed 2012–2020 and expected in 2020.** Countries are ordered according to the relative change in immigration during 2020; 95% CIs are included with the forecast.

index summarizing the joint restrictiveness level of all these measures. As described in the method section, we re-scaled original values to a 0–1 range to enable comparisons across variables. Fig 3 displays a heatmap with countries sorted according to the extent of immigration decline (X-axis), and the various stringency measures and changes in unemployment (Y-axis). Larger and darker circles indicate greater levels of stringency and rises in unemployment.

The results indicate that countries with higher overall levels of stringency experienced large (over 35%) or medium (20–30%) immigration declines; though, this relationship is not linear. Norway displays a relatively large drop in immigration despite moderate stringency levels, while Italy experienced a similar decline in immigration but recorded the highest levels of stringency. Restrictions on population movements seem to underpin these patterns. Travel, mobility and stay-at-home restrictions tend to display the highest levels of stringency in countries that report large or medium immigration declines. Australia, for instance, scored the highest levels of travel and mobility restrictions, and the largest decline in immigration. Yet, again, the degree of stringency does not seem to be the only factor determining the extent of immigration decline. Italy, Canada and the United States experienced similar levels of restrictions to Australia, but lower reductions in immigration. Rises in unemployment may have played a role in reducing levels of immigration in the United States and, to a lesser extent, in Canada, while they seem less prominent in other countries. Work and school closing do not show a clear relationship with cross-country variations in immigration.

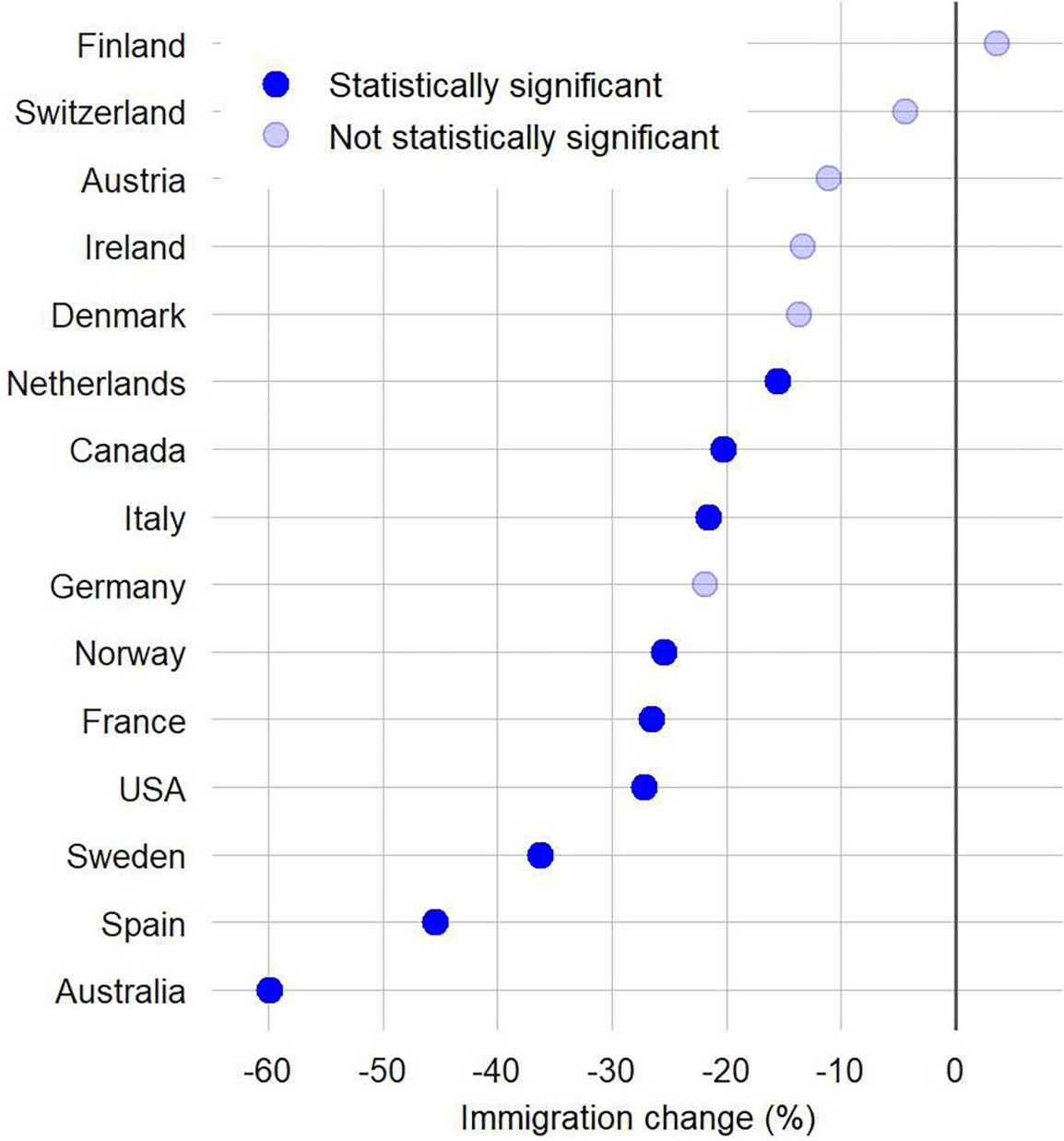

**Fig 2. Immigration changes between expected and observed values in 2020.** Countries are ordered according to the relative change in immigration during 2020; we consider changes in immigration as statistically significant when the observed immigration counts are outside the 95% CI of their respective forecast.

## Discussion and conclusion

Results show reductions in immigration for most countries in our sample during 2020, except in Finland. Declines ranged from 5% in Switzerland to 60% in Australia but drops below 15% appeared not to be statistically significant, probably due to the high uncertainty levels of the forecasted immigration counts. Immigration trends are volatile and hard to predict due to complex and interrelated drives both in origin and destination [23, 24]. Countries with declines in immigration higher than 20% reported more severe travel and mobility restrictions

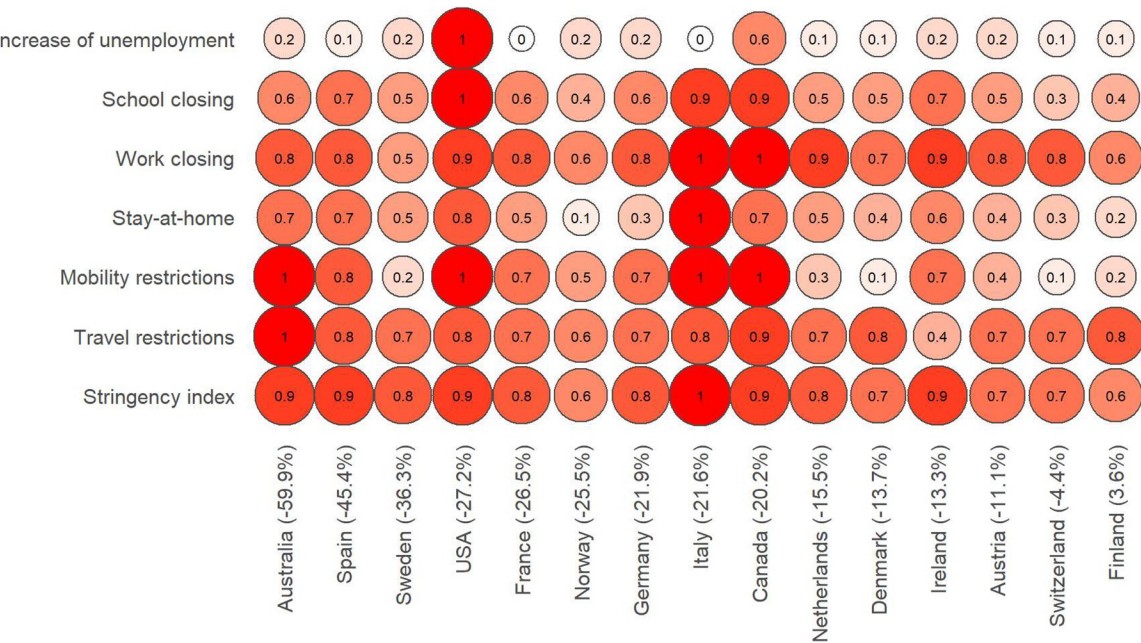

**Fig 3. Severity of stringency measures in 2020 (scale 0 to 1).** Countries are ordered according to the relative change in immigration during 2020, represented in brackets.

and stay-at-home requirements. However, the association is not linear, as countries with similar levels of stringency showed varying extents of immigration decline. Increasing unemployment, work and school closings do not show a relationship with cross-national variations in immigration.

Differences in the content of stringency measures, rather than differences in their restrictiveness level, may explain the varying reductions in immigration. Australia, for instance, maintained strict restrictions to international travel from all countries during 2020, while the European Union (EU) and other states from the European Free Trade Association (EFTA) gradually relaxed travel restrictions for EU+EFTA citizens but maintained certain restrictions for other countries. Thus, EU+EFTA states which typically receive a large share of immigrants from the EU+EFTA territory (e.g., Finland) may have been less affected by immigration declines, while European countries with large proportions of immigrants from outside EU +EFTA (e.g., Spain) may have been more impacted. In addition, most inflows to Spain come by airplane from overseas, specifically from Latin America, and this mode of travel could have been more affected by border restriction. Otherwise, we only consider stringency measures and travel restrictions in receiving countries. Different stringency measures and pandemic impacts amongst sending countries, such as the lack of financial resources to travel due to the economic hardship caused by the pandemic, could have impacted levels of emigration at origin and, thus, immigration in destination countries as well.

Our paper contributes some first empirical evidence of the extent of immigration decline drawing on a global sample of countries. Yet, the long-term impacts of COVID-19 on international migration are to be established. As travel restrictions are lifted, the risk of COVID-19 mortality decreases, and business activity resumes, immigration may rebound to pre-pandemic levels. However, recently released data from national institutes of statistics show that immigration may have returned to pre-pandemic levels in the Netherlands during 2021 but remained at low levels in Spain, suggesting cross-national variations in immigration recovery.

As foreign populations have become a pillar of ageing societies, monitoring immigration flows is key to guiding migration policy. A one-year reduction may not have a significant effect on accelerating population aging. But, if significant levels of reductions continue to occur, the demographic and economic sustainability of aging countries will be negatively impacted.

As data for subsequent years become available, future work could expand our analysis to explore the prevalence and durability of declines in immigration across countries. Future research could also seek to understand the underlying factors explaining variations in the extent of immigration changes and how they come about by analyzing the complexity of origin-destination migration flows. Analysis specific bilateral migration corridors can yield important insights and help to better interpret results because some specific corridors may have been more impacted than others. However, such analysis would be possible only for a few countries due to limited data. Analyzing origin-destination flows can enrich understanding of potentially transformative effects that COVID-19 may be having on the international migration system. Understanding these transformative effects is important given the fundamental role that migration plays in redistributing labor and skills to areas where they are needed [25], but also as a strategy to promote human development.

## Supporting information

**S1 Appendix.**
(DOCX)

## Author Contributions

**Conceptualization:** Miguel González-Leonardo, Michaela Potančoková, Francisco Rowe.

**Data curation:** Miguel González-Leonardo.

**Formal analysis:** Miguel González-Leonardo, Dilek Yildiz, Francisco Rowe.

**Funding acquisition:** Michaela Potančoková, Dilek Yildiz.

**Investigation:** Miguel González-Leonardo, Michaela Potančoková, Francisco Rowe.

**Methodology:** Miguel González-Leonardo, Dilek Yildiz, Francisco Rowe.

**Project administration:** Miguel González-Leonardo.

**Resources:** Miguel González-Leonardo.

**Software:** Miguel González-Leonardo, Dilek Yildiz, Francisco Rowe.

**Supervision:** Miguel González-Leonardo, Michaela Potančoková, Francisco Rowe.

**Validation:** Miguel González-Leonardo, Michaela Potančoková, Dilek Yildiz, Francisco Rowe.

**Visualization:** Miguel González-Leonardo, Francisco Rowe.

**Writing – original draft:** Miguel González-Leonardo, Michaela Potančoková, Dilek Yildiz, Francisco Rowe.

**Writing – review & editing:** Miguel González-Leonardo, Michaela Potančoková, Francisco Rowe.

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
