## [Decision Letter · Decision Letter 0]

14 Nov 2022

PONE-D-22-22645Quantifying the impact of COVID-19 on immigration in receiving high-income countriesPLOS ONE

Dear Dr. Miguel González-Leonardo,

Thank you for submitting your manuscript to PLOS ONE. After careful consideration, we feel that it has merit but does not fully meet PLOS ONE’s publication criteria as it currently stands. Therefore, we invite you to submit a revised version of the manuscript that addresses the points raised during the review process.

We look forward to receiving your revised manuscript.

Kind regards,

Ricky Chee Jiun Chia

Academic Editor

PLOS ONE

Journal Requirements:

“This analysis was supported by European Union’s Horizon 2020 research and innovation programme projects QUANTMIG (870299) & FUME (870649)”

“This analysis was supported by European Union’s Horizon 2020 research and innovation programme projects QUANTMIG (870299) & FUME (870649).”

Reviewers' comments:

Reviewer's Responses to Questions

**Comments to the Author**

1. Is the manuscript technically sound, and do the data support the conclusions?

Reviewer #1: Partly

Reviewer #2: Yes

2. Has the statistical analysis been performed appropriately and rigorously? 

Reviewer #1: No

Reviewer #2: Yes

3. Have the authors made all data underlying the findings in their manuscript fully available?

Reviewer #1: No

Reviewer #2: Yes

4. Is the manuscript presented in an intelligible fashion and written in standard English?

Reviewer #1: No

Reviewer #2: Yes

5. Review Comments to the Author

Reviewer #1: All comments and suggestions on the current manuscript are available in this submission's attached peer review report.

The Authors should take all comments and suggestions into consideration.

This paper needs a major revision and modifications to be accepted for publication in PLOS ONE.

Reviewer #2: Dear Authors,

The topic of the paper is very interesting and very well presented. The analysis is very well conducted and the results are very interesting. However, there are some points I would like to point out.

1) There is a lack of literature review, it would be interested to present excessively previous research on this topic which could provide more insights and explanations about your main aim. In addition, by providing literature review you can present more clearly the main scientific gap that you are trying to cover in this paper.

2) I am not sure why methods and materials are placed as a 4th part. I would suggest moving this part above the results. When writing a scientific paper it is better to have the methodology above the results as it makes it easier for the reader to understand the process you followed on your paper.

3) On you apendix "supporting information" you present a lot of valuable information that should be placed inside the main text as it can complete parts such as methodology, results and conclusions.

4) It would be beneficial to add a part with limitations of your research, future research, and applications of your results.

5) Please put a title on your figures and tables

Thank you.

6. PLOS authors have the option to publish the peer review history of their article (what does this mean?). If published, this will include your full peer review and any attached files.

Reviewer #1: **Yes: **Asst. Prof. Dr. Nemer Badwan / Assistant Professor of Economics and Finance

Reviewer #2: No

---

## [Author Response · Author response to Decision Letter 0]

22 Dec 2022

Dear Editor and reviewers,

Thank you for the opportunity to revise the manuscript and for your feedback. Please find responses to your comments in the attached documents.

Best wishes.

---

## [Editor Report · Decision Letter 1]

27 Dec 2022

Quantifying the impact of COVID-19 on immigration in receiving high-income countries

PONE-D-22-22645R1

Dear Dr. Miguel González-Leonardo,

We’re pleased to inform you that your manuscript has been judged scientifically suitable for publication and will be formally accepted for publication once it meets all outstanding technical requirements.

Kind regards,

Ricky Chee Jiun Chia

Academic Editor

PLOS ONE
---

## [Editor Report · Acceptance letter]

8 Jan 2023

PONE-D-22-22645R1 

Quantifying the impact of COVID-19 on immigration in receiving high-income countries 

Dear Dr. González-Leonardo:

I'm pleased to inform you that your manuscript has been deemed suitable for publication in PLOS ONE. Congratulations! Your manuscript is now with our production department. 

Kind regards, 

on behalf of

Dr. Ricky Chee Jiun Chia 

Academic Editor

PLOS ONE